

# Accounting for Uncertainties in Forecasting Tropical Cyclone-Induced Compound Flooding

Kees Nederhoff[1,2,5], Maarten van Ormondt[1], Jay Veeramony[3], Ap van Dongeren[2,4], José A. Á. Antolínez[5], Tim Leijnse[4,6], Dano Roelvink[2,4]

[1] Deltares USA, 8601 Georgia Ave, Silver Spring, MD 20910, USA
[2] CURR, UNESCO-IHE Institute for Water Education, P.O. BOX 3015, 2601 DA Delft, The Netherlands
[3] Naval Research Lab, Stennis Space Center, MS 39529, USA
[4] Marine and Coastal Management, Deltares, Boussinesqweg 1, Delft, 2629 HV, The Netherlands
[5] Delft University of Technology, Stevinweg 1, 2628 CN Delft, The Netherlands
[6] Institute for Environmental Studies (IVM), Vrije Universiteit Amsterdam, De Boelelaan 1111, 1081 HV Amsterdam, The Netherlands.

*Correspondence to*: Kees Nederhoff (kees.nederhoff@deltares-usa.us)

**Abstract.** Tropical cyclone impacts can have devastating effects on the population, infrastructure, and on natural habitats. However, predicting these impacts is difficult due to the inherent uncertainties in the storm track and intensity. In addition, due to computational constraints, both the relevant ocean physics and the uncertainties in meteorological forcing are only partly accounted for. This paper presents a new method, called the Tropical Cyclone Forecasting Framework (TC-FF), to probabilistically forecast compound flooding induced by tropical cyclones, considering uncertainties in track, forward speed, and wind speed/intensity. The open-source method accounts for all major relevant physical drivers, including tide, surge, and rainfall, and considers TC uncertainties through Gaussian error distributions and autoregressive techniques. The tool creates temporally and spatially varying wind fields to force a computationally efficient compound flood model, allowing for the computation of probabilistic wind and flood hazard maps for any oceanic basin in the world, as it does not require detailed information on the distribution of historical errors. A comparison of TC-FF and JTWC operational ensembles, both based on DeMaria et al. (2009), revealed minor differences of <10%, suggesting that TC-FF can be employed as an alternative, for example, in data-scarce environments. The method was applied to Cyclone Idai in Mozambique. The underlying physical model showed reliable skill in terms of tidal propagation, reproducing the storm surge generation during landfall and flooding near the city of Beira (success index of 0.59). The method was successfully applied to forecast the impact of Idai with different lead times. The case study analyzed needed at least 200 ensemble members to get reliable water levels and flood results three days before landfall (<1% flood probability error and <20 cm sampling errors). Results showed the sensitivity of forecasting, especially with increasing lead times, highlighting the importance of accounting for cyclone variability in decision-making and risk management.



## 1    Introduction

Tropical cyclone (TC) induced compound flooding, which occurs when storm surge, heavy rainfall, high tide, and river discharge coincide, can have devastating impacts on coastal communities (Wahl et al., 2015). This type of flooding is particularly concerning as it can result in higher water levels and increased inland flooding, leading to damage and loss of life (e.g., Resio & Irish, 2015). The increased frequency and severity of compound flooding events are expected to worsen due to the effects of climate change (e.g., Lin et al. 2012; Mori and Shimura, 2023) as well as on-going coastal development and population growth. Mitigation and preparedness strategies require a sound toolbox for assessing TC-induced compound flooding on coastal communities that enhance short to long-term decision-making.

Operational and strategic risk analyses are instrumental in analyzing and mitigating potential environmental risks. Operational risk analysis, typically associated with short-term forecasting (~several days), provides immediate response and preparedness for imminent disasters, ensuring the safety and protection of people and property (Roy & Kovordányi, 2012). Conversely, strategic risk analysis focuses on long-term climate variability assessments, delivering insights into hazards and their socio-economic and environmental impacts, thus facilitating informed policy decisions and adaptation strategies (e.g., Nederhoff et al., 2021). Though distinctly different, both perspectives are critical for comprehensive climate risk management, as they offer different scales and timeframes for prevention, preparedness, response, and recovery.

Forecasting agencies such as the National Hurricane Center (NHC) have significantly improved operational meteorological risk analysis, credited to gains made in numerical weather prediction models (McAdie et al., 2000, Cangialosi et al., 2020). Despite advancements, operational forecast errors remain significant enough to necessitate considering the inherent uncertainties in these forecasts for informed preparedness decision-making (Lamers et al., 2023). A common probabilistic approach is to represent the resulting uncertainty in track prediction by a cone envelope as a graphical representation that illustrates the possible track variation of the TC center (NHC, 2023). The shape of the cone can be derived from the historical error data of the forecast and typically represents a 66.7% probability that the track will be within the cone (i.e., 33.3% chance the track falls outside the cone). The cone increases in size with lead time as the errors in the prediction accumulate. While the cone gives valuable insight into the potential range of TC variability of the core, it can be easily misinterpreted as the corresponding impacted area, which can be substantially larger. Quantification of the uncertainty in track prediction can be computed with several methods. For example, De Maria et al. (2009) introduced a Monte Carlo method to generate 1,000 realizations by randomly sampling from historical error distribution functions from the past 5 yr for both the track and intensity. De Maria et al. (2013) improved their method so that the track uncertainty is estimated on a case-by-case basis using the Goerss predicted consensus error (GPCE; Goerss, 2007), where the uncertainty is estimated based on the spread of a dynamical model ensemble instead of historical averages. Other methods exist. For example, Chen et al. (2023) introduced a deep-learning ensemble approach for predicting tropical cyclone rapid intensification. However, these methods were all derived to provide





insight, before landfall, into the uncertainty of the wind speeds and not designed to force hydrodynamic or wave models, and
can thus result in too erratic forcing conditions.
Early Warning Systems (EWS) for coastal compound flooding are sensitive to uncertainties in the TC, including nonlinear
interactions between the TC size, forward speed, location of landfall, tides, rainfall, and infiltration. However, often EWS for
coastal flooding use physics-based and, due to computational constraints, deterministic approaches in which the best track is
used to force a hydrological & hydrodynamic model that computes the storm surge and the complex interactions between
coastal, fluvial, and pluvial processes (e.g., Global Storm Surge Information System; GLOSSIS based on Delft3D Flexible
Mesh; Kernkamp et al., 2011 and Coastal Emergency Risks Assessment; CERA based on ADCIRC; Luettich et al., 1992).
Several examples of probabilistic coastal flood methods capture uncertainty in forcing. For example, the Global Flood
Awareness System (GloFAS; Alfieri et al., 2013) is a modeling chain run by the European Centre for Medium-Range Weather
Forecasts (ECMWF) based on the LISFLOOD hydrological model forced by 51 ensemble members. While GloFAS is an
excellent resource for communities worldwide, it operates at a large scale with a relatively coarse resolution of 0.1 degrees
(~10 km), and is thus not designed explicitly for TCs that require high spatial resolutions (Roberts et al., 2020), and neither
account for relevant coastal processes such as tides. Higher resolution and the inclusion of coastal processes can be found in
several regional applications. For example, the Stevens Flood Advisory System (SFAS; Ayyad et al., 2022) is an ensemble-
based probabilistic forecasting of tide, surge, and riverine flow across the US Mid-Atlantic and Northeast coastline and runs
for 96 different atmospheric forcing datasets. Other examples include forecasting systems from the UK Met Office (Flowerdew
et al., 2010) and the Royal Netherlands Meteorological Institute (de Vries, 2009). All these systems rely on coarser numerical
forecasting products, focus on mid-latitude regions, and are thus not explicitly designed to forecasts hazards related to TCs.
Probabilistic modeling systems for TC-induced coastal flooding for operational risk analyses in the US and Japan include P-
Surge (Taylor and Glahn, 2008; Gonzalez and Taylor, 2021), which uses data from the NHC to create a set of synthetic storms
by perturbing the storm's position, size, and intensity based on past errors of the advisories. Subsequently, the Sea, Lake,
Overland, Surge from Hurricanes model (SLOSH; Jelesnianski 1992) is run and forecasts storm surge in real-time when a
hurricane is threatening. However, SLOSH does not account for several relevant (coastal) processes (e.g., tides, waves, rainfall,
infiltration) and thus lack their interactions. The Japan Meteorological Agency (JMA) does use a dynamic tide and storm surge
model (Higaki et al., 2009) but only accounts for a limited number of 11 ensemble members (Hasegawa et al., 2015). Moreover,
both methods are created with a specific region in mind and are not easily transferable to other locations.
Besides probabilistic physics-based techniques, statistical machine-learning techniques (e.g., Lecacheux et al., 2021 or Nguyen
& Chen, 2020) are becoming increasingly popular in reducing the computational expense of forecasting compound flooding.



However, these machine learning downscaling methods lack nonlinear interactions between relevant coastal processes driving
compound flooding. Hybrid methods focus on reducing the number of tracks simulated and proved capable of accurately
representing a larger set of scenarios (Bakker et al., 2022).

As introduced by Suh et al. (2015), the constraints in real-time forecasting for operational risk analysis are around both
'accuracy' and 'promptness'. In other words, the time constraints associated with forecasting dictate that some modeling systems
use a purely deterministic approach or a limited number of ensemble members to perform more detailed compound flooding
predictions and thus simplify the meteorological uncertainty (e.g., GLOSSIS, CERA, JMA). On the other hand, probabilistic
approaches for meteorology with a large number of ensemble members use simplified hydrodynamics or have an insufficient
resolution for TCs and thus do not account for the processes needed to forecast TC-induced coastal compound flooding (e.g.,
GloFAS, SFAS, NHC).

To address the limitations listed, we propose a method to generate probabilistic wind and compound flood hazard maps by
using, for the first time, ensembling techniques via statistical emulation of TCs combined with physics-driven modeling for
coastal compound flooding. The workflow emulates the TC evolution using an autoregressive technique in combination with
reported mean errors in track and intensity, similar to DeMaria et al. (2009) but without the need for historical error distribution
functions. Next, this emulator produces an ensemble of several (herein thousands) TC members. Then, for each ensemble
member, a time- and spatially-varying wind field is generated and used to force a computationally efficient compound flood
model SFINCS (Leijnse et al. 2021). The output consists of probabilistic wind and flood hazard maps that can be forecast on
time with limited computational resources anywhere in the world. This paper refers to the TC forecasting framework as the
Tropical Cyclone Forecasting Framework, TC-FF.

The paper is structured as follows. Section 2 introduces the Monte Carlo forecasting methodology. Section 3 describes the
case study site and historical event of interest. The materials and methods used in this paper are described in Section 4.
Validation in terms of tides and storms and application of the forecasting methodology are presented in Section 5. Finally,
Sections 6 and 7 discuss and summarize the main conclusions of the study.
**2   Tropical Cyclone Forecasting Framework**
In this paper, we introduce the probabilistic Tropical Cyclone Forecasting Framework, TC-FF, to compute TC-induced
compound flooding for operational risk analysis. Our approach integrates a TC emulator using a Monte Carlo-based ensemble
sampling generation with an autoregressive technique, which is a simplified adaptation of DeMaria et al. (2009). The ensemble
members are generated around the forecasted official track, considering the average historical errors in intensity, cross-track,
and along-track. Tthe ensemble members are provided as input for the fast compound flood model called SFINCS.





Additionally, TC-FF considers tidal movements, storm surge, precipitation, and infiltration. The outcomes are consolidated
into a unified probability product. By choice, each member has an equal likelihood of occurrence. The Python code for this
method is accessible on GitHub via the following link: https://github.com/Deltares/CoastalHazardsToolkit.

## 2.1 TC-FF flowchart

A compact flowchart of TC-FF used to generate the ensemble member is shown in Figure 1. The steps of this process are as
follows:

1. **Define settings**: The user specifies the data source, period, time step of the ensemble generation, and the number of ensemble members requested.

2. **Input best track:** The code either determines the best track based on gridded time and spatial-varying wind and pressure fields (e.g., COAMPS-TC; Doyle et al., 2014) or reads in the forecasted track by one of the forecasting centers (e.g., NHC or other agencies).

3. **Error matrices for along-track, cross-track, and intensity**: The tool first computes random realizations based on the along-track, cross-track, and intensity standard deviations imposed for the time steps requested. The imposed mean absolute error is scaled with the timestep to overcome any time step dependency.

4. **Generate ensemble members:** Following the approach of DeMaria et al. (2009), a Monte Carlo method generates numerous ensemble members based on error matrices of the previous step in combination with an autoregressive technique for the along-track, cross-track, and intensity error.

5. **Generate wind, pressure, and rain fields:** Generate meteorological forcing conditions, i.e., the surface wind and pressure fields per time step per ensemble member, based on parametric methods (e.g., Holland et al., 2010) for subsequent analysis and application within numerical models. Rainfall can be included as well via intensity relationships.

6. **Simulation and post-processing:** In this study, the compound flood model SFINCS is applied, but in principle, other hydrodynamic models can also be applied, albeit typically at a higher computational expense. Data from the different ensembles are combined into several probabilistic outputs ranging from the probability of gale force winds (wind speed > 35 knots or >18 m/s), compound flooding (water depth> 15 cm) to quantile estimates (e.g., 1% exceedance water level).





**Figure 1. Flowchart of the Tropical Cyclone Forecasting Framework (TC-FF). Pre-processing stages are represented in light blue, the computational core of ensemble generation is denoted in orange, the parametric wind field generation is portrayed in green, the hydrodynamic simulation and analysis of winds are marked in purple and outcomes in red.**





In the subsequent paragraphs, we describe in more detail the pre-processing, the computation of the ensemble members (track
and intensity variations), and the determination of time- and spatially-varying wind fields.

## 2.2 Pre-processing and input data

The pre-processing of TC-FF comprises three components.

First, one specifies the period they would like to simulate, including the total time period over which wind fields need to be
generated and the time period over which the ensembles need to be generated. In addition, a timestep for ensemble generation
(default 3 hours) needs to be specified. At this stage, one also specifies the mean absolute error and auto-regression coefficients
for the along-track, cross-track, and intensity. When these values are unknown, calibration needs to be performed to determine
them by comparing them with the reported errors of the forecast center (see calibration in Section 5.2.1). At this stage, one
also specifies the number of ensemble members requested. The influence of the number of ensemble members is discussed in
Section 5.3.2.

Second, since TC-FF creates random realizations around the best track, an input track is needed. Depending on the application,
TC-FF reads a forecast bulletin that generates the track or determine the best track from the output of a high-resolution regional
meteorological model. The determination of a track from a meteorological model is based on an algorithm that finds the
minimum pressure in an area of interest. It takes in grid values, u and v wind components, pressure, minimum distance for
clustering, and returns lists of x and y coordinates of cyclone eyes, the maximum wind speed plus pressure around each eye.

Third, before the generation of the ensemble members, TC-FF creates random errors with a normal distribution based on the
provided average errors. Matrices are two-dimensional, with one dimension being the number of time stamps and the other the
number of ensemble members. The imposed mean absolute error is scaled with the timestep to overcome any time step
dependency and converted into a standard deviation.

## 2.3 Ensemble members

### 2.3.1 Track realizations and calibration

An important component in TC-FF is the generation of track realizations (or ensemble members) from the official track
forecast. The official positions are interpolated with a spline function to include values at all requested times. Our approach
for the track realization largely follows DeMaria et al. (2009). We decompose the track error into the along-track (AT) and
cross-track (CT) components and account for the track error serial correlation via autoregressive regression (Equations 1 and
190 2).

$$AT_t = a_t AT_{t-i} + B_{rnd} \qquad\qquad \text{Equation 1}$$



$$CT_t = c_t CT_{t-i} + D_{rnd} \qquad \text{Equation 2}$$

in which $AT_t$ and $CT_t$ are the AT and CT error at time steps t, $a_t$ and $c_t$ are constants, $AT_{t-3}$ and $CT_{t-3}$ are errors of the previous
time step (typically i=3 hours), and B and D are random numbers that are normal (Gaussian) distributed, scaled with the mean
absolute error but are limited to +/- 2σ.

Unlike DeMaria et al. (2009), we do not access the probability distributions of historical errors. Instead, we calibrate the
parameters ($a_t$, $c_t$, and mean absolute errors for B and D) based on the reported historical errors from the agency responsible
for the issued forecast (see Section 5.2.1). This is a simpler methodology and requires substantially less data (which is also
typically not accessible outside the forecast centers). These historical errors are routinely reported by the forecast centers (e.g.,
see Section 4.1.2 for information on the data sources used in this paper). Note that errors in our implementation (neither the
error nor the auto-regressive coefficient) vary with lead time. We calibrate a constant mean absolute error in combination with
a single auto-regression coefficient (see Section 5.2.1 for calibration and Section 5.2.2 for the influence of simplifications).
Moreover, the mean absolute error is converted into a standard deviation using a fixed relationship assuming a normal
distribution of the error and scaled with the applied time step to allow the user flexibility in the applied time step.

The determination of the ensemble members is subsequently based on the sum of the forecast and random error components.
In other words, we add the along-track and cross-track error to forecasted along- and cross-track. An example of the first 20
ensemble members is presented in Figure 2B. Using this procedure, 10,000 ensembles are generated for each forecast case
within this study; however, it is possible to use fewer ensemble members to reduce the computational cost but at larger
statistical uncertainty (see Section 5.3.2 for trade-offs).
**2.3.2   Intensity realizations and calibration**
Similar to the track realization, the maximum wind speed (intensity) at a specific interval is determined using a random
sampling approach. The starting point is the official forecast of intensity that is interpolated to include values at all requested
times, and a random error component ($VE_t$) is added.

$$VE_t = e_t VT_{t-3} + F_{rnd} \qquad \text{Equation 3}$$

in which $VE_t$ at time steps t, $e_t$ is a constant, $VE_{t-3}$ are errors of the previous time step (typically 3 hours) and F is random
numbers that are normally distributed, scaled with the mean absolute error and is limited to +/- 2σ.

The inland wind decay model adjusts the maximum intensity as a function of the distance inland, is directly based on DeMaria
et al. (2009) and is computed with Equation 4. If the intensity of any inland ensemble member exceeds this predetermined
value at any forecast time, the intensity is adjusted to match this value. Subsequently, the intensity errors are recalculated based
on the adjusted intensity. Additionally, if the intensity of an inland ensemble member falls <7.7 m/s (15 knots) at any point in



time, the TC intensity is reset to zero for all subsequent periods to overcome any unrealistic reintensifying TCs. All these
criteria follow DeMaria et al. (2009).

$$V_i = 20 + 120e^{0.0035D} \hspace{3cm} \text{Equation 4}$$

in which the maximum wind speed ($V_i$) in knots and the distance to land (D) in kilometers (with negative values indicating
inland cyclones) are given, the intensity of an inland cyclone can be determined.

The intensity implementation differs from DeMaria et al. (2009) in the following ways. We remove the dependency that the
error scales with wind intensity and bias correction. Again, the determination of the ensemble members is based on the sum of
the forecasted and random components computed with Gaussian mean absolute errors and an auto-regressive constant over
lead time. Similar to the track realization, intensity errors are scaled with the time step to overcome any time step dependency.
The influence of the simplifications and the difference compared to NOAA operational code based on the original DeMaria et
al. (2009) and DeMaria et al. (2013) implementation are discussed in Section 5.2.2.
**2.4    Parametric wind fields**
After the determination of the ensemble members, the time and spatial varying wind fields are constructed and written in a
polar coordinate system . Several (horizontal) parametric wind profiles have been presented in the literature (e.g., Fujita, 1952;
Chavas et al., 2015), with the original Holland wind profile (Holland, 1980) being the most widely used due to its relative
simplicity. Several codes have been developed for storm surge models to provide time and spatial wind and pressure fields
(e.g., Hu et al., 2012 for ADCIRC). Deltares has developed the Wind Enhance Scheme (WES; Deltares, 2018) to generate TC
wind and pressure field around the specified location of a tropical cyclone center and given a number of TC parameters. In its
current implementation, information on wind radii (radius of gale force winds) can be considered in the Holland et al., (2010)
formulation using information either from best track-data or from the proposed relationships of Nederhoff et al. (2019), which
increases the accuracy of the method. Furthermore, the asymmetry of the wind field in a TC is also implemented, as delineated
by Schwerdt et al. (1979). Additionally, tropical cyclone-induced precipitation can be incorporated using empirical
relationships such as IPET (2006).
**2.5    SFINCS simulation and post-processing**
After the determination of the wind fields for all the requested ensemble members, TC-FF runs a hydrodynamic model. In this
study, we apply the compound flood model SFINCS (Leijnse et al., 2021), which lends itself well to a large number of
simulations in a reasonable amount of time due to its reduced complexity. SFINCS reads the tidal boundary conditions and
wind, pressure, and rainfall conditions from the wind fields. Once all the ensemble member simulations have finished,
probability products regarding wind and flood hazards are created. These products are created by sorting the results for each





grid cell and providing estimates for either specific intervals (e.g. wind speeds > 35 knots or water depth> 15 cm) or quantile
estimates (e.g., 1% exceedance water level).

## 3    Case study

The TC forecasting framework is applied to a historical event that took place in Mozambique's Sofala province: Cyclone Idai,
in March 2019. Mozambique is a country located in southeastern Africa (Figure 2A). The country has a diverse population of
over 31 million people, of which 2 million live in the Sofala province in central Mozambique (National Institute of Statistics
of Mozambique, 2017). Sofala is primarily rural, with small communities along the Pungwe and Buzi river deltas (Emerton et
al., 2020). Beira is the province's largest city, home to over 500,000 people, and an important port linking the hinterland to the
Indian Ocean. The city is prone to flooding, particularly during the rainy season, which generally extends from October to
April or May. This period coincides with the cyclone season, as cyclones often bring intense rainfall to the region. The
vulnerability of Beira to flooding is exacerbated by factors such as climate change, rapid urbanization, and limited
infrastructure.

Cyclone Idai was an example of a compound flood event that affected large parts of the coastal delta of Sofala (Eilander et al.,
2022). The storm began as a tropical depression in the Mozambique Channel, causing extensive flooding after its first landfall
in early March. It later intensified as it moved back over the sea, developing into a tropical cyclone with 10-minute sustained
wind speeds of 165 km/h. Idai made landfall near the port city of Beira, bringing powerful winds, resulting storm and heavy
rains that caused widespread flooding and destruction. Large areas were flooded, first around the coast and a few days later,
more inland in the Buzi and Pungwe floodplains. The total rainfall across the five days from March 13-18 ranged from 250–
660 mm (NASA GPM, 2019). Over 112,000 houses were destroyed, and an estimated 1.85 million people were affected (UN
OCHA, 2019).





**Figure 2. View of the study site: (A)** Mozambique's Sofala Province is situated in the southeastern region of Africa in the Southern Hemisphere. **(B)** Geographical and hydrodynamic representation of the study area. The SFINCS model extent, highlighted in Panel B, encompasses a portion of the Sofala region, forced offshore with a water level boundary, and is validated at seven tidal stations (indicated by orange circles; see Section 8.1). The best track is represented by a solid dark line, with the first 20 ensemble members 5 days before landfall demonstrated as gray lines. **(C)** The area of interest is the Pungwe estuary, situated near the city of Beira. Model validation also takes place at two high-water-marks close to the city (signified by a purple box), with model outcomes depicted at three diverse locations across the estuary (marked by circles).





# 4    Material and methods

## 4.1    Materials

### 4.1.1    Elevation datasets

Several topographic and bathymetric datasets were collected and combined to develop a merged DEM. Data includes field survey data points collected during three campaigns in November-December 2020 across Beira, locally-collected LiDAR, bathymetric charts, MERIT (Yamazaki et al., 2017) and GEBCO19 (IOC, IHO and BODC, 2003). Careful consideration was given to prioritize specific datasets in space to ensure the most detailed, recent, and accurate datasets were used in a given area. For example, survey and LiDAR data is bare earth and prioritized over the usage of MERIT and GEBCO19. The merged DEM was produced on medium-resolution (50 m) regional DEM, and a fine-resolution (5 m) local DEM in Beira. For more information on merging the data, one is referred to Deltares (2021).

### 4.1.2    Forcing conditions

Tidal boundary conditions were based on harmonic constituents provided by TPXO 8.0 (Egbert and Erofeeva, 2002), and tidal amplitudes and phases for all available 13 components were applied. The best track data (BTD) by the Joint Typhoon Warning Center (JTWC) is used throughout this study for meteorological forcing conditions (JTWC, 2022). Reported error statistics by the JTWC for the 5-year average from 2016-2020 were used to inform the ensemble generation (JTWC, 2021). Ensemble members from TC-FF were compared to 1,000 members produced with the code from NOAA, NHC, and JTWC based DeMaria et al. (2009) and DeMaria et al. (2013) that is used operationally (Buck Sampson, personal communication; June 5, 2023).

### 4.1.3    Validation data

Observed tidal coefficients near the city of Beira were used for the calibration and validation of the model (van Ormondt, 2020; see Figure 2 for locations). The validation of the event Cyclone Idai (2019) consisted of comparing both, observed and modeled flood extent in deltas of the Pungwe and Buzi rivers and high-water marks in the city of Beira. The observed flood extent was derived from Sentinel-1 synthetic aperture radar data (Eilander et al., 2022); and two observed high water marks (Deltares, 2021) were used, one at Praia Nova, in the western side of the city, and another one at the open coast beach in the southeast (see Figure 2 for locations). Correspondingly, values of modeled flood extent and high water marks were output at the same locations.



## 4.2 Methods

### 4.2.1 Area schematization

For this study, we employed the Super-Fast INundation of CoastS (SFINCS) model, which solves the simplified equations of mass and momentum for overland flow in two dimensions (Leijnse et al. 2021). The goal was to create one continuous compound flood area model that computes tidal propagation, storm surge, pluvial and fluvial flooding.

The area schematization builds upon Eilander et al. (2022) but varied in three ways. First, we extended the model alongshore and in deeper water to alleviate the need to nest in a large-scale regional coastal circulation model and generate tidal propagation and storm surge within the domain. Using a quadtree implementation (e.g., Liang et al. ,2008), we applied a variable model resolution ranging from 8000 to 500 meters. A quadtree is a technique in which the refinement from one level to another is based on the original cell but divided into 4 smaller cells with 2 times smaller grid size and allows extending the model setup into deeper water without having time step restrictions in deeper water based on the explicit numerical scheme of SFINCS. Second, high-resolution topo-bathymetry and land roughness were included in the native resolution utilizing sub-grid lookup tables (Leijnse et al., 2020). However, the hydrodynamic computations were performed on a coarser resolution to save computational time. Up to 10-meter DEM information was included in the 500-meter grid cells (i.e., factor 50 refinement). Lastly, sub-grid bathymetry features were included to account for maximum dune height based on the DEM to control overflow during storm conditions around Beira.

A spatially-varying roughness and infiltration was used based on land elevation. All points above mean sea level (MSL) have a high Manning friction coefficient of 0.06 $s/m^{1/3}$, and an infiltration rate of 1.9 mm/hr (typical values from HSGs Group C; United States Department of Agriculture, 2009), and all other points have lower friction of 0.02 $s/m^{1/3}$ to represent water and do not have any infiltration. The SFINCS model was forced with tidal boundary conditions and time- and spatially-varying winds, pressure, and rainfall fields. We refer to Section 8.1 for calibration of the tides, in which we show that the area model reproduces tides with a median MAE of 21 cm. Wind and pressure fields were created with the Holland wind profile (Holland et al. 2010) based on the BTD (see Section 2.4 for details). Rainfall for TCs was based on the Interagency Performance Evaluation Task Force Rainfall Analysis (IPET, 2006) method. Comparison with the reported rainfall total revealed a significant underestimation of cumulative rainfall during Idai based on IPET. Based on the magnitude of the underestimation, rainfall estimates by IPET were tripled, resulting in a cumulative rainfall in the area of interest of 495 mm for the best track, which is in a similar order of magnitude as observed (see Section 3).

### 4.2.2 Simulations periods

The validation of the area schematization focused on two time periods. First, 3 spring-neap cycles (January 13, 2022, until February 26, 2022) were used for the tidal calibration and validation in the area of interest (see Appendix 8.1). Second, Idai





was hindcasted forced with the JTWC BTD and compared to observational data for flood extent and high-water levels (Section
5.1). After validation of the area schematization, the new forecasting methodology introduced in Section 2 was applied. Various
lead times ranging from 1 to 5 days before the second landfall for 10,000 ensemble members were computed (Section 5.3).

Model runs were performed on the Deltares Netherlands Linux-based High-Performance Computing platform using 10 Intel
Xeon CPU E3-1276 v3. On average, a 7-day Idai simulation took about 4 minutes on a single core. Running all 50,000 events
took ~15 days.

### 4.2.3    xModel skill

Several accuracy metrics were calculated throughout this study: model bias, mean-absolute-error (MAE; Equation 5), root-
mean-square-error (RMSE; Equation 6), unbiased RMSE (uRMSE; RMSE with bias removed from the predicted value). These
error metrics are used for comparison in water levels, wind speed and track errors.

$$MAE = \frac{1}{N}\sum(|y_i - x_i|) \qquad \text{Equation 5}$$

$$RMSE = \sqrt{\frac{1}{N}\sum(y_i - x_i)^2} \qquad \text{Equation 6}$$

where N is the number of data points, $y_i$ is the i-th prediction (modeled) value, $x_i$ is the i-th measurement.

Moreover, skill is quantified by binary flood metrics (Wing et al. 2017). The model output (M) is converted to one of two
states: wet (1) or dry (0), using a commonly used threshold of 15 cm (e.g., Wing et al. 2017) and compared to the Sentinel
benchmark data (B). The Critical Success Index (C; Equation 7) accounts for both overprediction and underprediction and can
range from 0 (no match between modeled and benchmark data) to 1 (perfect match between modeled and benchmark data).

$$C = \frac{M_1 B_1}{M_1 B_1 + M_0 B_{1+} M_1 B_0} \qquad \text{Equation 7}$$


For the comparison of cumulative distribution functions (CDF) of cross-track, along-track and intensity, we also applied the
Continuous Ranked Probability Score (CRPS; Matheson & Winkler, 1976). CRPS measures how good forecasts are in
matching observed outcomes; where CRPS = 0, the forecast is wholly accurate, and CRPS = 1, the forecast is wholly
inaccurate.





$$\text{CRPS}(\mathbf{F}, \mathbf{x}) = \int_{-\infty}^{\infty} [F(y) - F_0(y)]^2 dy$$

Equation 8

where F(y) is the CDF is associated with an empirical probabilistic reference and prediction.
**4.2.4    Analysis method**
The analysis of forecasting results was undertaken using several methods. Initially, extreme wind speeds and water levels were
assessed by charting them as time-series data, inclusive of quantile estimates such as the 95% confidence interval (CI).
Following this, the maximum values registered during the simulation were organized into cumulative distribution functions
(CDFs). This process offered insights into their exceedance probability. Finally, the mean probability of flooding was
computed. The method to derive this value entailed counting the instances where computational cells registered a minimum of
15 cm of water. Only cells positioned above mean sea level (MSL) were incorporated into the area estimates.



## 5    Results

### 5.1    Verification of the numerical model for Cyclone Idai

Computed water levels near Beira show the strong tidal modulation and the wind-induced storm surge during the landfall of the cyclone (Figure 3 – panel A; blue line for water level and vertical line for moment of landfall). Based on the difference between the predicted astronomic tide and the total modeled water level, we estimate a storm surge of >3.5m due to the ~45 m/s wind speeds (Figure 3 – panel B). The storm surge at Beira is driven by wind setup as well as pluvial and fluvial drivers. Deeper in the estuary, in the Pungwe flood plains, water levels peaked several days after landfall due to intense upstream rainfall and subsequent runoff. Water levels near Buzi Village seem to be a combined result of first marine and second riverine-driven water levels.

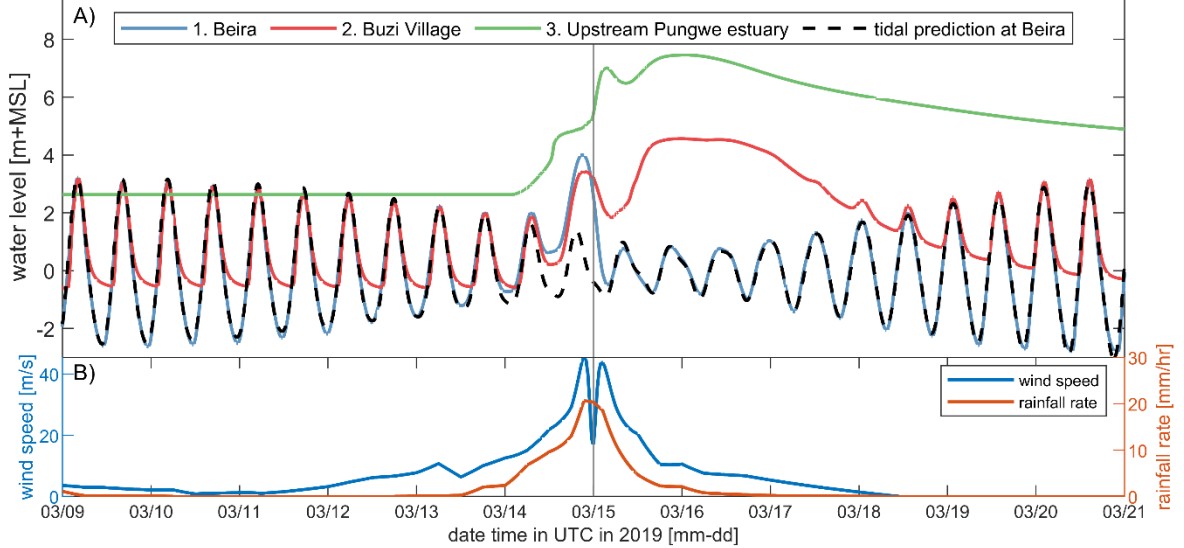

**Figure 3. Time series of water levels, wind speed, and precipitation within the study area. (A) Computed water levels at various locations (blue for Beira, red for Buzi village, and green for upstream in the Pungwe estuary (see Figure 2C for their location) and the black dashed line representing the astronomical prediction at Beira. (B) Simulated wind speed (blue) and rainfall rate (red) over the same period. Idai made landfall on March 15, and its powerful winds and rainfall resulted in marine flooding at Beira and riverine-driven flooding upstream in the estuary. The vertical line represents the moment of landfall.**

Validation of the SFINCS model for the observed extent (blue colors in Figure 4A) gives confidence in the ability to simulate the compound flooding (Figure 4). The model can reproduce the Sentinel-1 derived extent with a Critical Success Index of 0.59. This skill score is comparable to previous work by Eilander et al. (2022), albeit somewhat lower. Based on the differences between the modeled and satellite-derived extent, it becomes apparent that the model underestimates the flooding around the Buzi River (false negative; orange colors in Figure 4B around 660-7800 km). We hypothesize this is due to the lack of river inflow related to an underestimation of rainfall further upstream and/or overestimation of infiltration due to soil saturation which is not considered. Moreover, the comparison with satellite-derived flood extent indicates an overestimation of the flooding at Beira (false positive; red colors in Figure 4). Here, we suspect that the benchmark data might be off, and the coastal





flooding already receded before the Sentinel data recorded the extent. The observed high-water marks near Beira ranged from
3.6m within the estuary to 2.9 m + MSL at the open coast and are reproduced by SFINCS with respectively 3.8 and 3 m +
MSL. This difference suggests a positive bias of the model results at the coast of ~10-20 cm, similar to the tidal validation (see
Appendix 8.1), which revealed a median MAE of 21 cm.

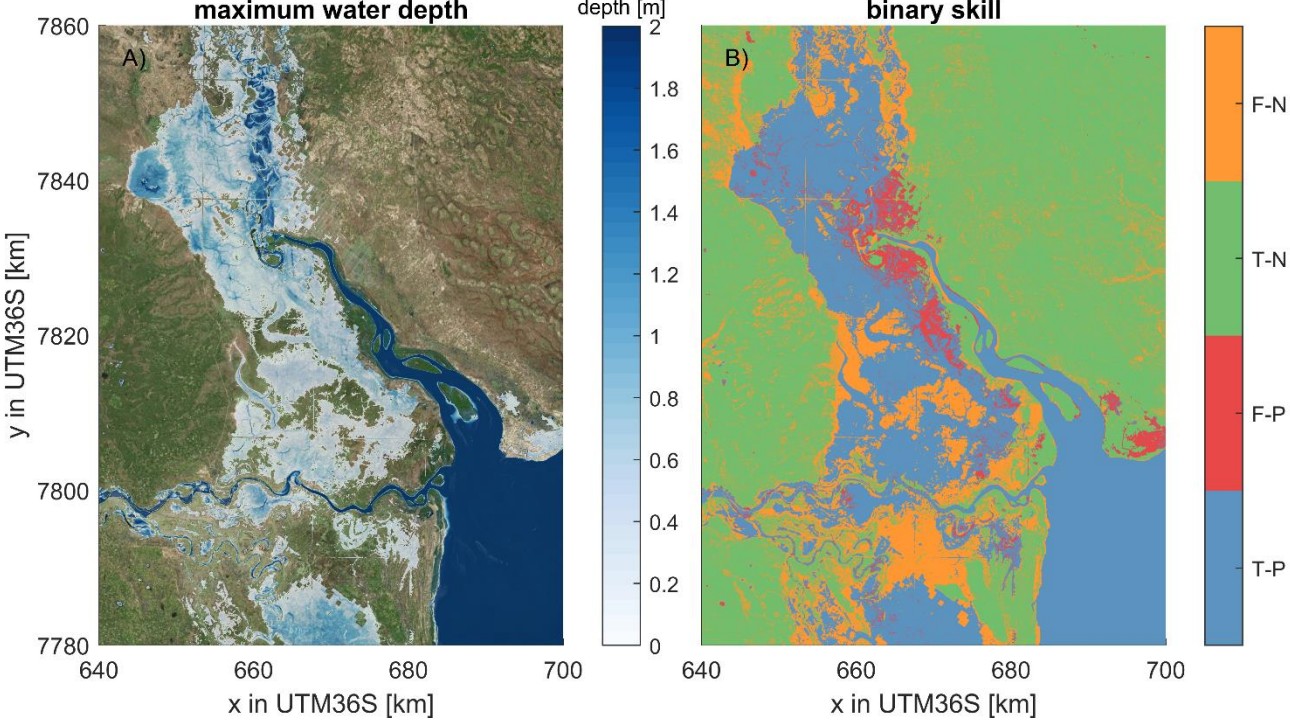


**Figure 4. Maximum computed water depth (Panel A) and binary skill of flood extents for Idai (Panel B). Water depths are downscaled from the model resolution to the 10x10 meter resolution of the topo-bathymetry. The binary skill evaluation (Panel B) assists in determining the model's accuracy and dependability, and the Sentinel-1 radar data is used as a reference to determine skill. A true-positive (T-P) outcome denotes a correct flood prediction by the model compared to Sentinel-1 derived extent, whereas a false-positive (F-P) occurs when the model forecasts a non-existent flood. In contrast, a false-negative (F-N) indicates where the model overlooks an actual flood, and a true-negative (T-N) result occurs when the model accurately predicts the lack of a flood event. The model produces large-scale flooding, which is largely also observed in the data, but local differences of over- and underestimation exist. The coordinate system of this figure is WGS 84 / UTM 36 S (EPSG 32736). © Microsoft.**

404



## 5.2 Calibration and influence of simplifications of TC-FF

### 5.2.1 Calibration of TC-FF: mean absolute error and auto-regression

This study used JTWC-reported errors for the along-track, cross-track, and intensity for the Southern Hemisphere to calibrate our methodology (JTWC, 2021). For other case studies, for example, based on different forecasting agencies or in other ocean basics), these reported errors can be used instead. Calibration is performed by minimizing the square-root difference between computed and reported mean absolute values for various lead times using the Nelder-Mead method. This effort resulted in mean absolute errors for B and D of 68.5 and 55.3 km and autoregression coefficients $a_t$, $c_t$, of 1.214 and 1.181 (Figure 5A and B) for the along-track and cross-track. Moreover, we calibrated the mean absolute error and regression coefficients for the intensity, which resulted in mean absolute errors for F of 9.28 m/s and autoregression coefficient $e_t$ of 0.624 (Figure 5C).

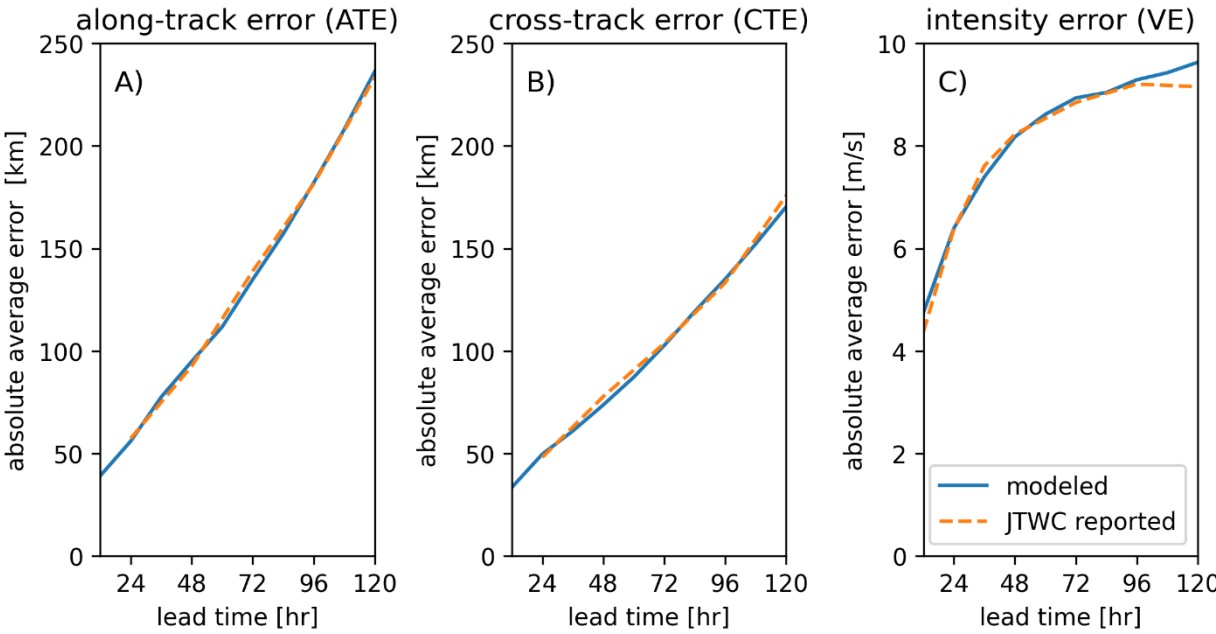

**Figure 5. Comparison of calibration results for the probabilistic forecasting method TC-FF (solid blue line) and the Joint Typhoon Warning Center (JTWC) reported error statistics based on the 5-year average (2016-2020) in the Southern Hemisphere (dashed orange line). Panel A represents the along-track error, Panel B demonstrates the cross-track error, and Panel C exhibits the wind speed or intensity error. Modeled errors are based on 1,000 ensemble members. Modeled absolute average errors are similar to JTWC.**

### 5.2.2 Comparison of TC-FF with operational forecast products

Errors produced by TC-FF are compared to the implementation from NOAA, NHC and JTWC that are used operationally. Minor differences between the TC-FF and full implementation based on DeMaria et al. (2009) and De Maria et al. (2013) exist and are attributed to the simplifications used in the error distribution (including the lack of GPCE) and lack of bias conditions. The distribution in along-track, cross-track, and intensity error is typically in the same order (Figure 6), which is confirmed by





a median CPRS over various lead times from 0 to 120 hours of 0.07, 0.05, 0.10 and median MAE of 37 km, 21 km, and 7 m/s
of for respectively the along-track, cross-track, and intensity. At the same time, TC-FF has by design no bias corrections,
whereas the operational system does, leading to the positive median along-track error in red compared to the blue line in Figure
6A and a median bias of -16 km. Besides the median estimates, the interquartile range (25-75%) and 95% CI  match relatively
well for the along-track and cross-track errors. Larger differences are found for the intensity error. In general, the wind intensity
error looks visually erratic and doesn't start at zero for no lead time, which is the result of the inland wind decay model. Both
JTWC and TC-FF have a negative bias due to the effect of land, but TC-FF does have a median bias of +6.7 m/s compared to
JTWC, suggesting that TC-FF overestimates. However, more substantial differences are found for the interquartile range and
95% CI. These findings for the along-track, cross-track, and intensity are supported by a more detailed analysis of the CDF
for the different parameters as a function of lead time (Figure 12, Figure 13, Figure 14). For the along-track and cross-track,
we observe an increase in the MAE and uRMSE as a function of lead time but a decrease in the CPRS. The increasingly larger
error distribution influences this pattern. Moreover, TC-FF produces Gaussian-distributed errors while the JTWC error
distribution differs since it is based on historical error distribution and adjusted based on the GPCE. Similar to Figure 6, larger
differences are found for the intensity error, which is influenced by the bias correction that increases with lead times.





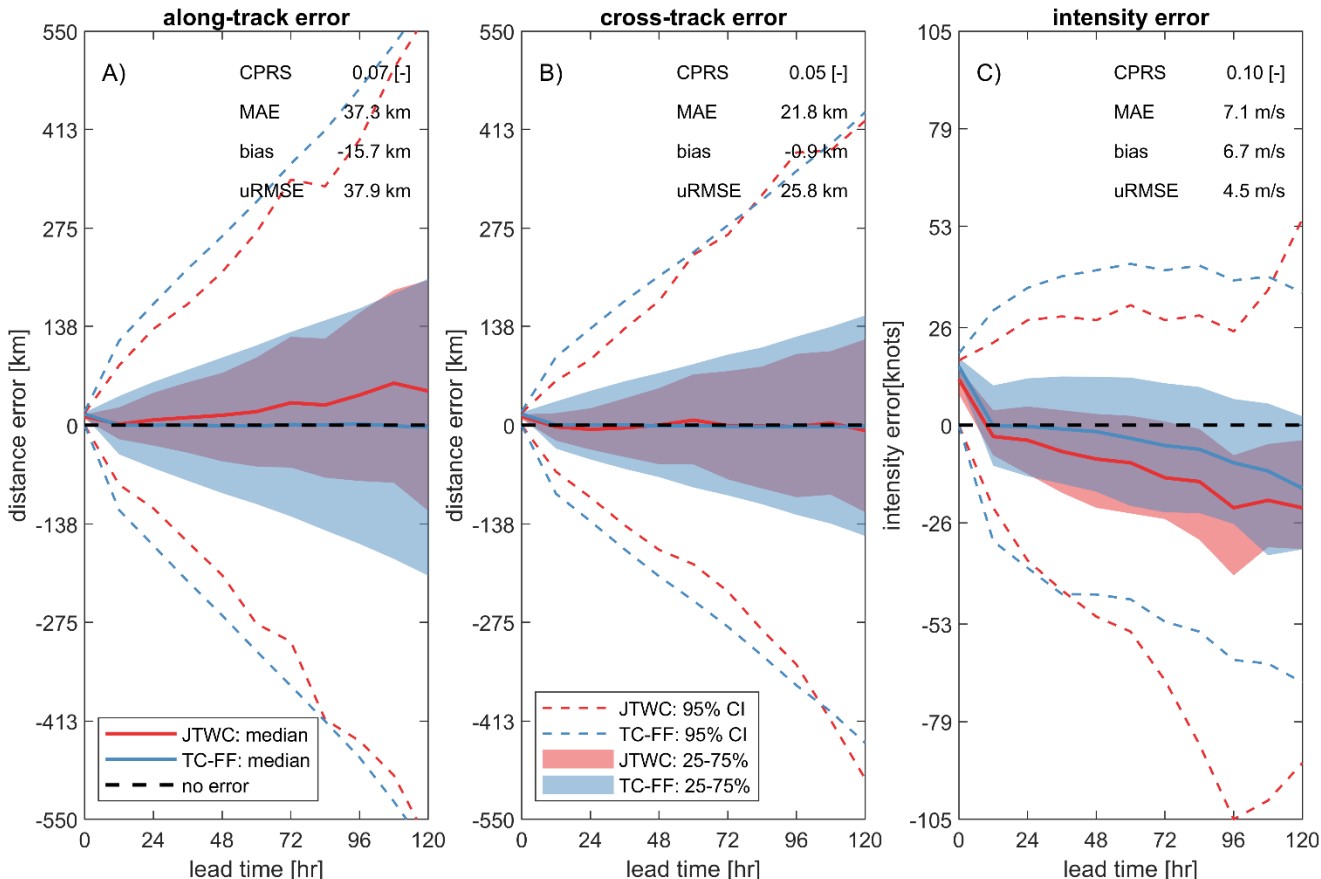

**Figure 6.** Comparison of validation results for the probabilistic forecasting method TC-FF (blue line) and the Joint Typhoon Warning Center (JTWC) operational product (red line). Panel A represents the along-track error, Panel B demonstrates the cross-track error, and Panel C exhibits the wind speed or intensity error. Errors are computation for both the TC-FF and JTWC are based on 1,000 ensemble members. Solid lines are median estimats, shaded areas the interquartile range (25-75% CI) and dashed line the 95% CI. TC-FF and JTWC produce broadly similar error distributions for different lead times.

## 5.3 Forecasting of Idai using the TC-FF

This section presents the application of forecasting Idai using the TC-FF.

### 5.3.1 Uncertainty three days before landfall

The TC-FF method with 10,000 ensemble members is applied to the case of Cyclone Idai. The results reveal that accounting for the uncertainty of the TC track and intensity of eye three days before landfall results in considerable uncertainty regarding wind speeds and water levels near Beira (Figure 7) or the region (Figure 8). In particular, the wind speeds show a 95% CI of about 7-40 m/s at the moment of landfall (Figure 7A) versus ~45 m/s or a Saffir-Simpson Hurricane Wind Scale (SSHWS) of 2 of the best track. Moreover, TC category 1 wind speeds could occur as early as March 15 at 21:30 UTC or as late as March



16 at 19:50 UTC. This spread of possible maximum wind speeds at Beira results from the large uncertainties in intensity and
a difference in landfall location and time. Based on the same model simulations, the empirical cumulative distribution function
(CDF) of the maximum wind speed at Beira ranges from 8.8 to 59.2 with a median wind speed of 25.5 m/s, while the best
track has a 5.9% exceedance probability (Figure 7B). Consequently, water levels vary greatly (Figure 7C). For example,
ensemble members can exhibit a sizeable wind-driven setup due to TC wind blowing from offshore into the estuary, pushing
water up in the estuary and at Beira. For landfall locations west of the estuary, the wind blows offshore, resulting in a large
set-down. Note that Beira is in the Southern Hemisphere, and due to the Coriolis effect, TCs spin clockwise. The highest water
levels occur when high tide and wind-driven setup coincide, which explains the three peaks in the 95% CI water level given
the semi-diurnal tide and the highest possible wind speed for ~1.5 days (Figure 7C). The maximum water levels are dominated
by the tide except in the situation of cyclone impact (see the CDF in Figure 7D and the minimum value of ~3.5m+MSL around
90%, which is influenced by the tide and time window over which it is determined). The specific track of Idai resulted in
relatively extreme conditions compared to other possible combinations (both for winds and water levels). A similar pattern
can be observed in the spatial maps shown in Figure 8. The average probability of flooding in the area is 26%, with higher
probabilities of flooding found in the lower-lying portions of the estuary (note we are excluding points below MSL; Figure
8A). The 1% exceedance flood depth threshold shows a large extent and is quite similar to the computed extent due to Idai
(see Figure 4A for comparison with Figure 8B). The main difference is that there is more flooding near the city of Beira and
somewhat less near Buzi Village. The match between the 1% exceedance flood depth and the best track with Idai suggests that
the event was relatively severe and implies that even though many other potential scenarios could have unfolded, they likely
would not have resulted in the same extensive flooding caused by Idai.



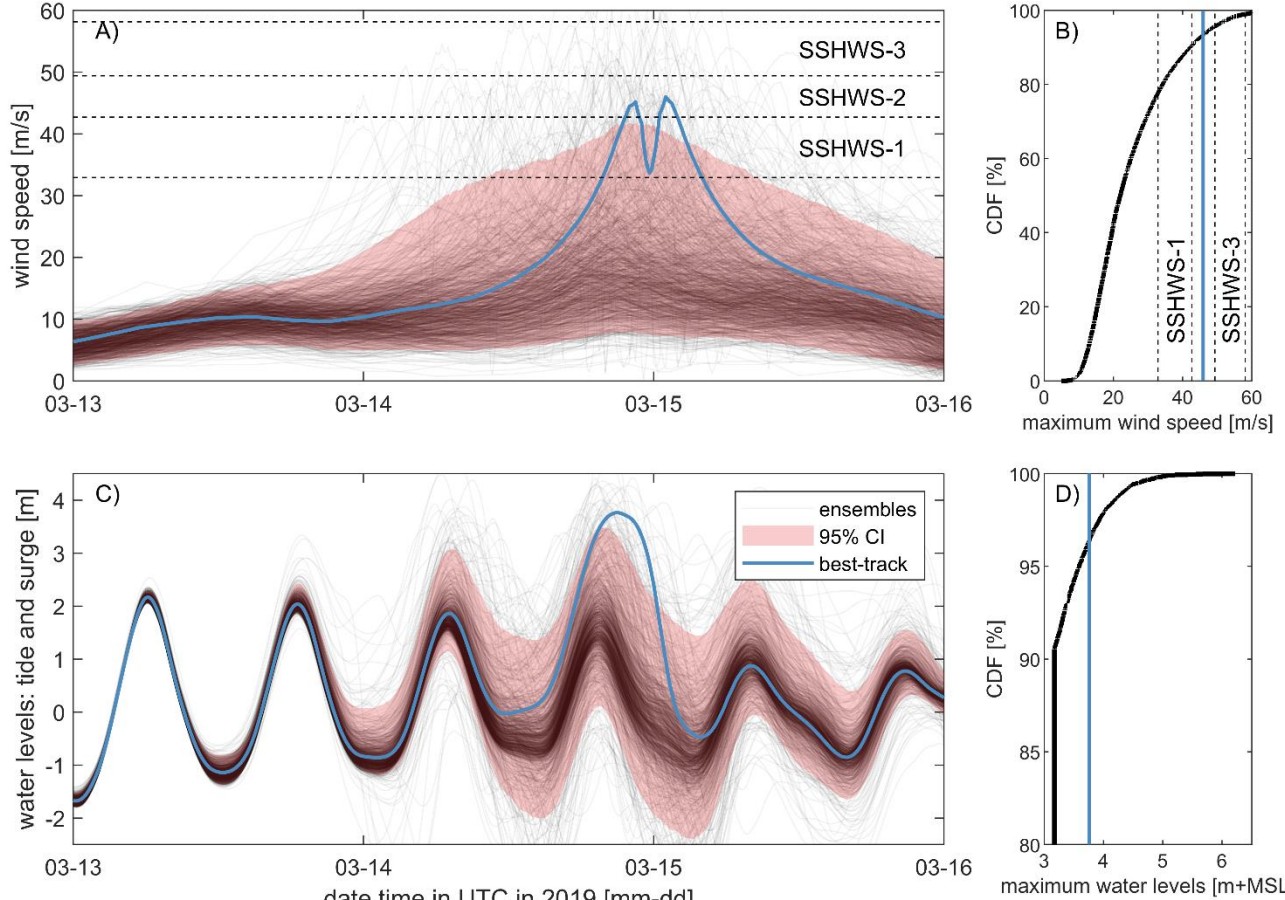

**Figure 7. Multi-panel Analysis of wind and water Levels three days before landfall: (A) time series of wind speeds, (B) maximum wind speeds, (C) time series of water levels near Beira, and (D) maximum water levels. Data is derived from 10,000 ensemble members (black transparent line; every 10[th] plotted) with red shading representing the 95% CI. The best track (blue line) and the Saffir-Simpson Hurricane Wind Scale are included for comparison (panels A and B only). There is substantial uncertainty in wind speeds and water levels near Beira three days prior to landfall.**



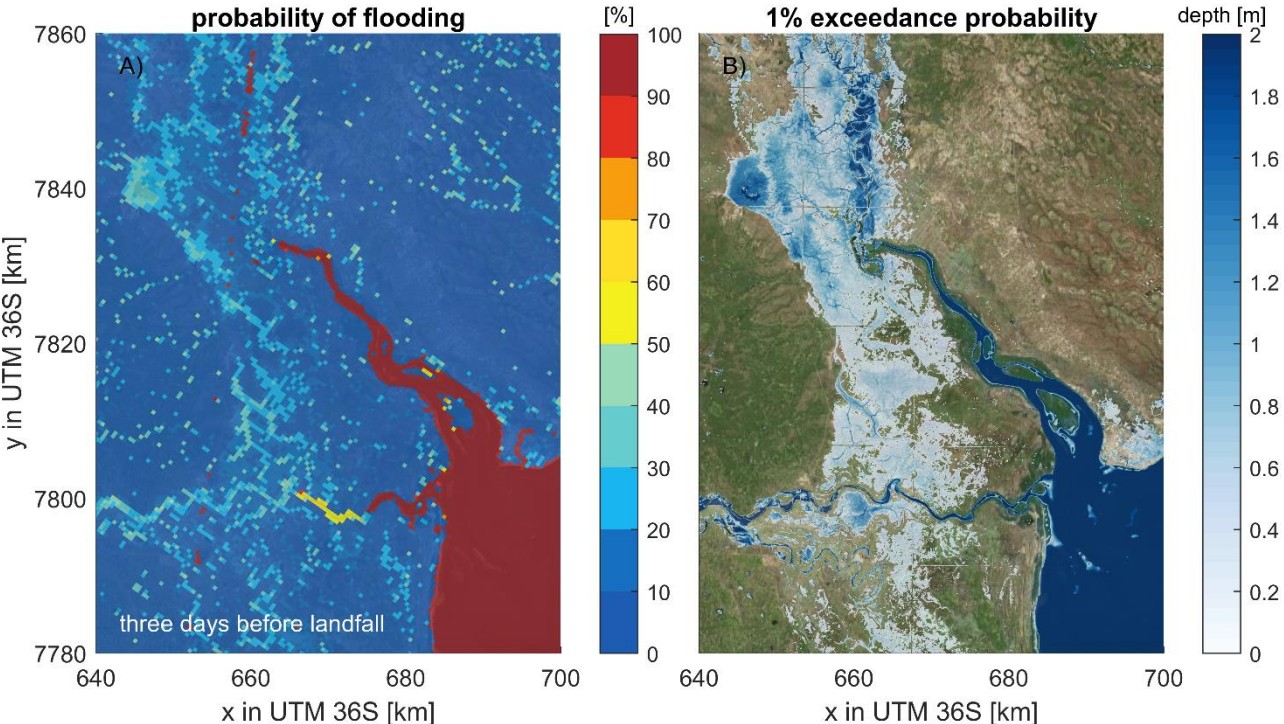

**Figure 8. Probabilistic flood analysis for Cyclone Idai three days before landfall: (A) Spatial distribution of flooding probability; (B) Corresponding 1% exceedance water depth estimates, highlighting areas at most significant hazard. Results in panel A are determined from 10,000 ensemble members on the original 500-meter model resolution, while water depth in panel B is downscaled to the original 10x10-meter bathymetry resolution. Higher probabilities of flooding are found in the lower-lying portions of the estuary. The coordinate system of this figure is WGS 84 / UTM 36 S (EPSG 32736). © Microsoft.**

### 5.3.2 Influence of sampling size

As described by Cashwell and Everett (1959) and DeMaria et al. (2009), the precision of Monte Carlo techniques is proportional to the number of ensemble members (N). The convergence rate typically shows a slower progression than 1/N, constituting a limitation intrinsic to all Monte Carlo methods. To investigate the convergence rate and the error induced by employing a finite number of ensemble members, the Idai forecasting case three days prior to landfall is used, analogous to the preceding section, albeit with a variable number of ensemble members. Additionally, bootstrapping is employed to approximate convergence rates and the accompanying uncertainty.

The estimation of the 95% exceedance maximum water levels in proximity to Beira exhibits convergence with the number of ensemble members, albeit with considerable deviations compared to a fully converged solution with 10,000 members when implementing a low number of ensemble members (Figure 9A). For instance, employing merely 50 ensemble members results in an interquartile range (25-75%) of -0.28 to +0.10 m. Increasing the number of ensemble members reduces this sampling uncertainty to a range of -0.09 to +0.06 m for 200 ensemble members.



Similarly, the standard deviation for several quantiles of maximum water level estimates at Beira reduces with more ensemble
members. It exhibits a similar pattern from higher to lower quantiles (Figure 9B). In essence, estimating rare events necessitates
executing more ensemble members to attain comparable convergence. This study found that the 95% exceedance maximum
water level at Beira when utilizing 200 ensemble members has a standard deviation of 21 cm (blue line Figure 9B). This level
of convergence seems acceptable since it is in a similar order as the skill of the hydrodynamic model (see Section 5.1).

The probability of error in flood potential is expressed as a function of N on a log-log plot (Figure 9C). Compared to a fully
converged solution with 10,000 members, for N=200, the mean error constitutes 0.95%, and the maximum error amounts to
1.53%. Note that this estimate is without considering the model error. In the log-log diagram, the errors exhibit near-linear
correlations with N and could serve as a basis for determining the number of ensemble members needed for a specified
confidence level. For instance, to achieve a maximum error of 1% in flood probability, it would be necessary to utilize 500
ensemble members.

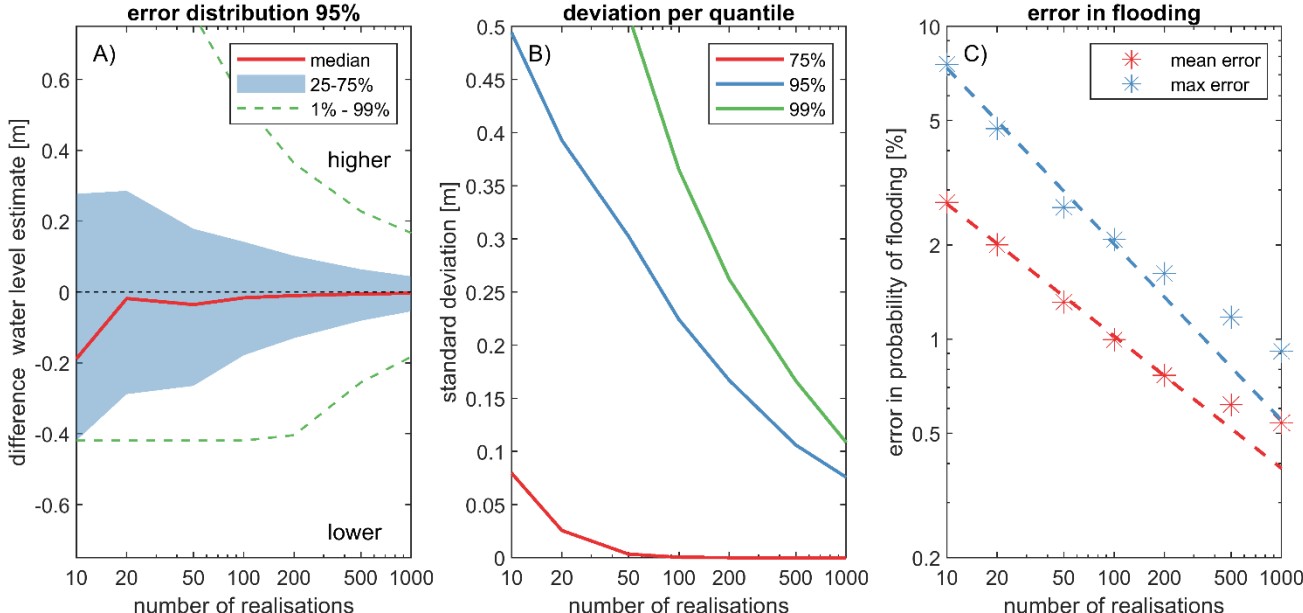


**Figure 9. Sampling size effects on flood estimation accuracy. (A) Quantiles of sampling error for the 5% exceedance water level. (B)**
**Standard deviation of 75%, 95%, and 99% quantiles, illustrating the uncertainty in estimation. (C) Comparison of maximum and**
**average error in flood probability predictions. All panels were generated using 10,000 ensemble members and a 1000-bootstrap**
**resampling approach. Using more ensemble members reduces the sampling uncertainty.**
**5.3.3    Importance of lead time**
Thus far, the probabilistic TC forecasting framework has been implemented three days prior to the landfall of Idai.
Nevertheless, the forecast's results fluctuate with lead times, consequently influencing the associated evaluations of water
levels (Figure 10) and flood probabilities (Figure 11).



The predicted water levels  (tide + surge) vary with lead times (Figure 10A and C). Specifically, at a lead time of five days before landfall, an (unsurprisingly) larger spread between the ensemble members is observed compared to lead times of, for example, one or three days. Moreover, as landfall approaches, the time series converges since increasing ensemble members produce highly similar predictions. For example, notice how individual ensemble members 1 day before landfall show similar storm surges and still water levels (i.e., the concentration of lines which becomes more apparent in Figure 10). Moreover, the 5% and 95% exceedance values become less spread out and more peaked around landfall (dashed lines in Figure 10). This convergence is more apparent for the storm surge. The CDF of the maximum storm surge levels increases with reducing lead time (Figure 10B). For example, the median storm surge increases from 0.5m five days before landfall to 0.9 and 2.0m for lead times of three days and one day, respectively (notice the increasing median estimate in the CDF plot from 5 to 1 day in Figure 10B). This increase in maximum storm surge shows the increasing certainty that the TC will land near Beira. However, for other locations, the opposite may occur as the landfall shifts away from it. The still water levels are influenced by both tidal motions and the influence of the TC (Figure 11C). This strongly influences the maximum computed still water level (Figure 11D). For instance, the lowest maximum water level for all simulations is around ~2 m above MSL, resulting from the maximum tidal range rather than the TC itself. The 95th quantile of the maximum still water level is 3.4 m + MSL five days prior to landfall, which increases to 3.6 and 4.0 m+MSL for lead times of three days and one day, respectively. The best track of Idai is included as a reference and estimated to have a 9% probability of exceedance 1 day before landfall.

A large portion of the Sofala province faces a minor flood risk five days before the actual landfall. The flood probability for the estuary near Beira increases as lead times reduce (Figure 11B). In particular, the average probability of flooding five days before landfall is 15%, increasing to 17 and 24% for lead times of three and one day, respectively. Conversely, for the entire model domain, a probability of greater than 1% flooding declines from 97 to 94 and 64 km$^2$ for lead times of five, three, and one day (Figure 10A). In other words, five days before landfall, less confidence in predictions translates into more spatial variability on flooding probability tied to a larger impact area. Closer to the actual landfall, there is more certainty over which area will be affected.



**Figure 10. Forecasted water levels in Beira for 1-5 day lead times: temporal evaluation and cumulative distribution. Panel A and C: Time series illustrating the forecasted water levels in proximity to Beira with lead times ranging from 1 to 5 days prior to landfall showcasing both individual ensemble members (solid transparent lines; every 100th plotted), tide-only (brown), best-track (black) and quantile estimates (95% dashed lines). Panels A and C use the same colors and line styles. Panel B and D: Cumulative distribution function (CDF) showing the maximum water levels in ascending order for all ensemble members, providing insights into the probability of occurrence for various water level thresholds. Panels B and D use the same colors. Panels A and B show the storm surge levels (computed still water levels minus predicted tidal levels), while Panels C and D present the still water level (tide and surge).**



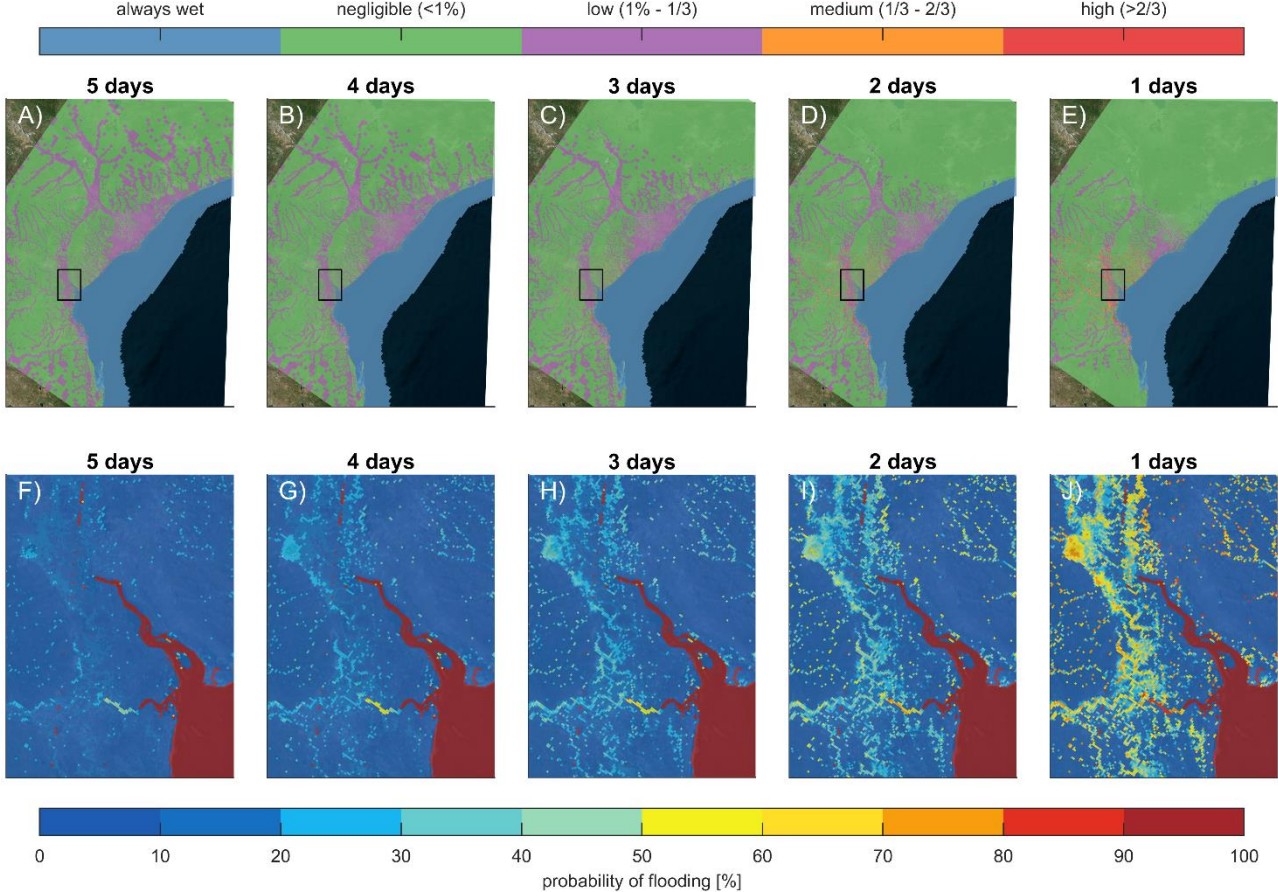

**Figure 11. Evolution of the flood probability prior to landfall: Panels A-E depict the spatial distribution of flooding probabilities at 5, 4, 3, 2, and 1 day(s) before landfall, respectively. Color gradients represent the varying probability. The top panels focus on the entire area simulated and the bottom panels on the Pungwe and Buzi river deltas. With decreasing lead time, the area that could be affected decreases while there is an increased probability of flooding near Beira. The coordinate system of this figure is WGS 84 / UTM 36 S (EPSG 32736). © Microsoft.**





## 6   Discussion

This paper describes a new probabilistic method to forecast TC-induced coastal compound flooding by tide, surge, and rainfall using Monte Carlo sampling. Due to the limited number of observations on TC evolution, for short-term operational analyses, an autoregressive technique that imposes potential errors on top of the forecasted track is preferred over those parametric sampling techniques used for long-term and strategic risk assessments based on historical records (e.g., Nederhoff et al., 2021). In addition, for the same scarcity of observation, there is limited knowledge of the underlying joint distribution between TC and ocean characteristics, which makes Monte Carlo sampling preferred compared to sampling techniques that are highly efficient for complex multivariate patterns such as cluster analysis (e.g., Choi et al., 2009) and MDA methods (e.g., Bakker et al., 2022). However, exploring the possibility of increasing efficiency via the aforementioned methods is important, especially since the error space increases as a function of lead time, and estimating these events requires increasing amounts of ensemble members (Figure 9B). However, this is a topic that requires an in-depth analysis and is beyond the scope of the present study.

Compared to the implementation of DeMaria et al. (2009) and DeMaria et al. (2013), the ensemble generation is simplified by removing bias corrections, applying a single normal error distribution calibrated on historical errors (Figure 5), and does not account for GPCE. While we acknowledge these simplifications, this method does make it possible to account for TC forecasting errors for any ocean basin based on reported average historical errors alone. Nevertheless, the behavior of a specific tropical cyclone (TC) does not necessarily conform to the "average" pattern, and differences between the operational JTWC model were found (Figure 6). The implications of these assumptions on the precision and predictive proficiency of our approach for coastal compound flooding remain undetermined. Our implementation has been recently integrated into an operational system tailored for the contiguous United States. Verification of the reliability of this operational system is currently pending. Regardless, TC-FF compares well with the predictions provided by ECMWF of Idai that showed a probability of 50 to 90% of severe flooding four to one day before landfall (Figure 10).

In the introduced methodology, we apply the compound flooding model SFINCS. While the validation gave confidence that the hydrodynamic model reproduces the main tidal motions and flooding during Idai, differences did exist compared to the data (Figure 4). The model skill could be improved by including additional wind radii information in the parametric wind model (e.g. radius of gale force winds along different quadrants) and more accurately resolving on-land winds, rainfall, and infiltration processes. For example, Done et al. (2020) present a methodology to account for terrain effects by adjusting winds from a parametric wind field model by using a numerical boundary layer model. Here, we applied the IPET empirical relationship that relates pressure drop to rainfall intensity. Initially, deployment showed the necessity to triple the rainfall rate due to severe underestimation of the total rainfall and associated flooding. Improvement (stochastic parametrization) of TC rainfall could overcome this limitation. Moreover, SFINCS was run with a constant infiltration rate and does not account for drainage systems and flood protection measures besides the frontal levee. It is also unknown how the topo-bathymetry that



was collected before Idai influenced results. Lastly, the effects of waves (e.g., setup, runup, overtopping) and morphological
change were not considered. All these limitations affect the model skill and could explain some mismatches observed compared
to Sentinel-1 data and high-water marks at Beira. However, the computational efficiency of SFINCS allowed us to run
thousands of ensemble members on limited computational resources. We accept the loss of some model accuracy with this
gain of speed.

The focus of the development of TC-FF has been geared to the computation of overland flooding. However, TCs pose
significant hazards through both water *and* wind. A study by Rappaport (2014) indicated that from 1963 to 2012 in the United
States, approximately 90% of fatalities associated with tropical cyclones were due to water-related incidents. The wind-related
fatalities were about 8%. This does not provide insight into the cause of damage associated with landfalling TCs, nor does it
provide insight into how these ratios vary across the globe. Regardless, TC-FF does provide the possibility to estimate extreme
wind speeds and link this to potential damage as an additional data product. Including wind damage as part of our framework
is something we are planning to work on in the future.

## 7   Conclusions

A new method and highly flexible open-source tool was developed to perform probabilistic forecasting of tropical cyclone-
induced coastal compound flooding. The Tropical Cyclone Forecasting Framework, TC-FF, computes a set of ensemble
members based on a simplified DeMaria et al. (2009) method. In particular, TC-FF uses gridded time- and spatially-varying
wind and pressure fields or forecasted tracks and combines this with historical observed error on the along-track, cross-track,
and intensity. Subsequently, the tool creates a temporally and spatially varying wind field, including rainfall, to force a
computationally efficient compound flood model. This approach allows for the inference of probabilistic wind and flood hazard
maps calibrated to any ocean basin in the world with limited computational resources. In contrast to the current practice, TC-
FF allows uncertainty analysis using large ensembles produced with physics-based models, narrowing down confidence bands
on forecasting coastal compound flooding focused on operational TC risk analyses.

The validation of the quadtree SFINCS model for Mozambique's Sofala province showed reliable skill in terms of tidal
propagation in the area of interest (median MAE of 21 cm), including good skill in reproducing the observed flood extent for
the case of the flooding caused by Cyclone Idai (2021). The model was able to reproduce the storm surge generation during
landfall and flooding near the city of Beira, including the subsequent compound flooding resulting from rainfall runoff in the
Pungwe estuary (critical success index of 0.59). Moreover, the model runs efficiently with a wall clock time of 4 minutes for
a 7-day event allowing it to be deployed in probabilistic operational assessments when using multiple cores.



TC-FF was calibrated with the average reported errors for the southern hemisphere via the Nelder-Mead method to determine
the mean absolute errors and autoregression coefficients. A comparison between TC-FF and JTWC (based on the complete
implementation of DeMaria et al., 2009) and DeMaria et al., 2013) revealed minor differences. In particular, for various lead
times from 0 to 120 hours, a median Continuous Ranked Probability Score (CRPS) of 0.07, 0.05, 0.10 and median MAE of 37
km, 21 km, and 7 m/s for respectively the along-track, cross-track, and intensity error were found. These findings give
confidence that the TC-FF, including the simplified DeMaria et al. (2009) implementation, can be used for more generalized
applications in data-scarce environments.

TC-FF provides valuable insights into the uncertainty of wind speeds, water levels, and potential flooding due to Idai, revealing
the impacts of track and intensity uncertainties. This is demonstrated in the wide array of possible maximum wind speeds and
significant fluctuations in water levels, which are primarily affected by tidal influences and the cyclone. For instance, even
just three days prior to landfall, there's a broad spread in the predicted flood areas. This suggests that there is still a significant
chance that Idai may not hit the anticipated area or may not generate a substantial storm surge.

The precision of forecasts is directly related to the number of ensemble members used. A mean error in flood probability of
less than 1% and <20 cm sampling errors for the 1% exceedance water level at Beira required  200 members. Based on that,
we determine that at least 200 ensemble members are needed to get reliable water levels and flood results three days before
landfall. A higher number of ensemble members reduces sampling uncertainty and increases the accuracy of water level and
flood potential estimates.

The lead time before landfall has a considerable impact on the forecast's precision. As the lead time decreases, the variability
of forecasts diminishes, and the forecasts converge to similar predictions. Similarly, the probability of flooding in certain areas,
such as the estuary near Beira, increases as the lead time shortens, providing more certainty over the areas that will be affected
by the event.

TC-FF offers a significant advancement compared to the current status quo of a single deterministic simulation when
forecasting tropical cyclone compound flooding hazards. This approach facilitates a comprehensive understanding of complex
interdependencies and uncertainties. By quantifying the likelihood of various outcomes, probabilistic methods enable
stakeholders to make more informed decisions, allocate resources better, and enhance preparedness and resilience in the face
of these catastrophic natural phenomena.

*Code and data availability.*



The code and data are freely available to other researchers and consultants. The Python code for this method is accessible on
GitHub via the following link: https://github.com/Deltares/CoastalHazardsToolkit.

*Author contributions.*
KN and MvO developed the method and the outline for the manuscript. KN wrote the initial manuscript, with editorial
comments by JV, AvD, JA, T and DR.

*Competing interests.*
The authors declare that they have no conflict of interest.

*Acknowledgments and financial support*
The authors thank the Deltares research programs 'Natural Hazards' and 'Risk Analysis and Management', which have provided
financing to develop and write this paper. Moreover, we want to thank Buck Sampson for input and data regarding the
operational wind field probabilities.



## 8    Appendices

### 8.1    Tidal calibration and validation

A tidal calibration was performed on the SFINCS computed tidal constituents compared to the tidal constituents at Beira. Constituents with an amplitude of more than 5 cm (M2, S2, N2, K2, and K1) were adjusted in terms of amplitude (multiplication) and phase (addition). Amplitude changes varied between 0.84 and 1.07 while phase difference changed on average by 40°. These calibration steps of adjusting the tidal constituents substantially reduced tidal errors at the Beira from a MAE of 43 to 17 cm. Secondly, model skill in reproducing tidal amplitudes and phases is assessed at 7 tide stations across the area of interest (including the calibration station of Beira). The SFINCS model reproduces tide with a median MAE of 21 cm, median RMSE of 25 cm, and median difference in M2 and S2 amplitude and phase of respectively -10 and -1 cm and -10 to -12° (median values computed over the different stations). Our hypothesis is that the reduction in tidal error observed at Beira throughout the calibration process might be due to a misalignment in the amplitudes and phases of the TPXO model which were used to generate the tidal boundary conditions (see Section 3.1.2). Presumably, the bathymetry contributes to the error observed in the validation process.

**Table A1. Evaluation of model proficiency in replicating tides near the Sofala province. Stations are ordered south to north. Columns one and two present the Mean Absolute Error (MAE) and Root Mean Square Error (RMSE), respectively, as error metrics for the comparison between observed and simulated tidal time series. The final four columns display the discrepancy (Δ) in amplitude (A) and phase difference (ϕ) for the two most prominent tidal constituents in the area (M2 and S2), where Δ is calculated as the difference between observed and simulated values.**

| Name | MAE [m] | RMSE [m] | ΔM2 A [m] | ΔM2 ϕ [°] | ΔS2  A [m] | ΔS2 ϕ [°] |
|---|---|---|---|---|---|---|
| Bazaruto | 0.13 | 0.15 | -0.10 | -7 | 0.01 | -2 |
| Bartolomeu Dias | 0.12 | 0.15 | -0.14 | 1 | -0.11 | -1 |
| Chiloane | 0.30 | 0.41 | 0.20 | -10 | 0.08 | -15 |
| Beira | 0.17 | 0.20 | 0.00 | 0 | 0.00 | 0 |
| Chinde | 0.21 | 0.25 | -0.08 | -13 | -0.01 | -12 |
| Quelimane | 0.26 | 0.32 | -0.14 | -15 | -0.09 | -21 |
| Pebane | 0.21 | 0.25 | -0.14 | -11 | -0.09 | -15 |
| Median | 0.21 | 0.25 | -0.10 | -10 | -0.01 | -12 |



Figure 12, Figure 13 and Figure 14 provide additional information for Section 5.2.2. 'Influence of simplification in TC-FF'.

**Figure 12. Comparison between the cumulative distribution function (CDF) of the along-track-error (ATE) for JTWC (red; reference) and TC-FF (blue; modeled). The different panels represent different lead times.**





**Figure 13. Comparison between the cumulative distribution function (CDF) of the cross-track-error (CTE) for JTWC (red; reference) and TC-FF (blue; modeled). The different panels represent different lead times.**





702

**Figure 14. Comparison between the cumulative distribution function (CDF) of the intensity error (VE) for JTWC (red; reference)**
**and TC-FF (blue; modeled). The different panels represent different lead times.**

705



706

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
