# Peer review of "Accounting for Uncertainties in Forecasting Tropical Cyclone-Induced Compound Flooding"

_EGUsphere, 2023_

## Author Comment (AC2)

**Rebuttal letter manuscript "Accounting for Uncertainties in Forecasting Tropical Cyclone-Induced Compound Flooding"**

Dear editor, dear reviewers,

On 11 October 2023, we submitted the manuscript "Accounting for Uncertainties in Forecasting Tropical Cyclone-Induced Compound Flooding" (MS No.: egusphere-2023-2341) to Geoscientific Model Development (GMD). On January 4, 2024, we were informed that the open discussion was completed. We received comments from three reviewers which provided positive and constructive feedback on the work done. We would like to acknowledge their time and efforts, which have led to an improvement in the quality of our manuscript. Below you find a point-by-point reply to all specific questions and suggestions. Attached you also find the revised manuscript with the changes made to address the review comments tracked. Moreover, we have addressed the comment from the Executive Editor focused on highlighting the issues regarding our compliance with the "Code and Data Policy" of Geoscientific Model Development on 26 December 2023.

Kind regards,

Kees Nederhoff
* * *
**Anonymous Referee #1 (04 Dec 2023)**

This paper presents a novel, flexible and open-source modelling framework, TC-FF, to probabilistically forecast tropical cyclone-induced compound flooding. The framework combines a Monte Carlo-based ensemble sampling generation with an autoregressive approach to create possible tropical cyclone scenarios, accounting for the uncertainties in track, intensity, and forward speed. The framework uses these scenarios to force a hydrodynamic model, SFINCS, that simulates the compound effects of tide, surge and rainfall on coastal flooding. The framework produces probabilistic flood maps that can support operational risk analysis and decision-making. The framework has been successfully applied to Cyclone Idai in Mozambique, and reproduced the tidal propagation, storm surge generation, and flooding extent near the city of Beira with reasonable model skills. The authors also analyze the sensitivity of the flood forecasting results to the number of ensemble members and the lead time.

This is a well-researched and nicely written academic paper. The paper is worth being published with some modifications addressing the comments below.
We want to thank Reviewer #1 for their kind words.

Detailed comments:

1. In the first paragraph of Introduction, please provide some examples to support the claim that compound flooding events are expected to worsen due to climate change and coastal development.
We have included some examples of how compound flooding is expected to worsen due to climate change and coastal development (L36-40).

2. In line 73, the paper should clearly state the disadvantages and limitations of the WES methods after analyzing their characteristics.

We presume the reviewer is referring to Early Warning Systems (EWS – L70-80) instead of the Wind Enhancement Scheme (WES; discussed in L250). In the revised manuscript, we have added the limitations of EWS focused on GLOSSIS and CERA (L74-80). We feel we already addressed the limitations of the other EWS in the manuscript (see L81-100).

3. The paper has logically explained the limitations of the current methods for forecasting tropical cyclone-induced compound flooding in the Introduction. However, the paper should add a summary of these limitations at the end of this section, and then state the research gap and research questions that motivate this study. The paper should also highlight the novelty and contribution of the proposed method, TC-FF, and how it addresses the limitations of the current methods. An explicit explanation of the advantages and benefits of TC-FF over other existing methods should be provided after line 108.

We agree with the reviewer and have added a summary of the limitations of the current methods (L115-119). Additionally, the novelty and contribution of our proposed method, TC-FF, are outlined throughout the manuscript, including the introduction (L121-129).

4. In line 310, the paper states that the compound flood area model computes tidal propagation, storm surge, pluvial and fluvial flooding. However, the paper does not provide much information on fluvial flooding. If fluvial flooding is included in the model, the paper should add the details of fluvial forcing conditions in the Material part. The paper should also explain how the authors obtain and process the fluvial data, such as the river discharge, the river network, and the boundary conditions.

Fluvial flooding is accounted for in the model via rain-on-grid computations in combination with infiltration estimates to simulate surface runoff and its subsequent river discharges which cause fluvial flooding. We have clarified this in the revised manuscript (L354-356) and included a discussion on the limitations of this approach (L636-638).

5. From line 312 to line 314, how did the authors extend the model alongshore and in deeper water. Is there any references or principles about defining the water level boundary. Please give details.

We added more background on how we extended the model alongshore and in deeper water (L328-330). Moreover, we added more background information on the principles regarding water level boundaries (L347-348).

6. In line 321, the paper mentions that the model uses sub-grid bathymetry features to account for the effects of dunes and channels on the water level and flood extent. However, the paper does not provide much information on how the authors obtain and apply the sub-grid bathymetry data, especially in the estuarine area where bathymetry data is very difficult to get. The paper should explain the source, resolution, and processing of the sub-grid bathymetry data, and how they are incorporated into the model.

We added more background information on how bathymetry was used (L336-341). For details on the source and resolution, we refer to Section 4.1.1 (L297-303).

7. In line 343, the paper states that the model runs on a high-performance computing (HPC) platform, but it does not specify the type of parallel computing used by the model. The paper should clearly indicate if the model uses parallel computing by CPU or GPU, or if it only runs on a single CPU core on the HPC platform.

We added a description of the HPC characteristics (L364-367).

8. In line 368, the part of "Result" is divided into three subsections: verification of the numerical model, calibration and validation of the ensemble generation, and probabilistic forecasting of compound flooding. I suggest adding a brief introduction at the beginning of the section for a summary.

We followed the suggestion and added a brief introduction in the results section (L392-397).

9. The paper validates the SFINCS model for Cyclone Idai by comparing the model results with a limited number of observations, such as two high-water marks and one flood extent map. However, these data sources may not be enough to evaluate the model performance in different locations and time periods. The paper should use more data sources, such as satellite imagery or field survey, to assess the model spatio-temporal accuracy and reliability in simulating the compound flooding event.

We agree with the reviewer that additional data sources would be very valuable for the validation of this globally applicable and generalizable model set-up and method developed. Here we focus on the development of the methodology with the application to one relevant case study. We have used all the available data including field survey data with the two high watermarks and satellite imagery with the Sentinel-1 derived extent. We clarified this in the revised manuscript (L622-625). The methodology presented here is robust and fast to carry out uncertainty or sensitivity analysis, however, there is a scarcity of training/validation data and additional data sources do not exist for this particular case study.

10. In the last paragraph, the paper briefly mentions that the results can be used for operational risk analysis, but it does not provide any explanations or examples of how the proposed model can support practical applications. The paper should add some explanations of how the results can be used for operational risk analysis, such as providing examples and recommendations for the use of the probabilistic flood maps.

We added an explanation of one of the possible practical applications focused on flooding in the conclusion section (L700-701). However, at the end of the day, it is up to risk managers to define their strategies and use the outcomes from tools such as TC-FF.

11. In line 129, there is a typo "Tthe", which should be "The".

Thank you for noticing and it has been corrected in the revised manuscript (L142).
* * *
**Anonymous Referee #2 (07 Dec 2023)**
This paper presents a new probabilistic forecasting framework for tropical cyclone (TC)-related compound flooding. The framework captures the influence of uncertain track location, TC wind speed (intensity), and storm forward speed. The framework is demonstrated for the case of Cyclone Idai.

The novelty of the framework is the integration of factors driving compound flooding. A key strength of the paper is the robust probabilistic verification, presented in crisp, accessible diagrams. The work is a substantial technical contribution and should be well received by the risk modeling community. While some of the modeling choices could be criticized for being too simple, and missing nuances in the wind and rainfall fields, I understand that complexity was a necessary sacrifice to ensure timely computation of many thousands of iterations.

The demonstration case includes insightful explorations of sensitivity. For example, understanding how widespread low risk sharpens up to more localized high risk as lead time shortens is an important result.

The system has global applicability, and I look forward to seeing how broadly applicable the presented results are to other regions. While the system is framed as an operational risk analysis tool I can see that the system will also enable research into the predictability of compound flooding. For example, the result that 200 ensemble members are needed for acceptable convergence 3 days out has implications for predictability studies.

The paper is very well written and organized and is a pleasure to read. The forecasting system is well motivated and situated within the context of existing systems. I also appreciate that the code is open source. However, the link to the python code was broken for me.

The subject matter is appropriate for GMD and is worth being published after my comments below have been addressed.

We thank Reviewer #2 for their time and kind words. We have addressed the broken link in the revised manuscript (https://github.com/Deltares-research/cht_cyclones) and also uploaded the source code to Zenedo including DOI https://doi.org/10.5281/zenodo.10433070.

Specific Comments

My expertise is in climate and tropical cyclone modeling so my specific comments come from that background.

My main comment is to ask how confident you are that this system samples the primary sources of uncertainty for compound flooding. The system is heavily weighted towards uncertainty in track parameters. Are these the main sources of uncertainty for compound flooding? How important is uncertainty in rainfall characteristics, for example? Even for the track parameters, how important is your decision to universally apply historical error statistics vs a flow-dependent error sampling. I would encourage some additional discussion about the level of confidence in the estimates of the true uncertainty, and how this may vary from case to case.

We want to thank Reviewer #2 for their insightful comment and frankly, we agree with this statement.

- TC-FF accounts for the uncertainties regarding the track but does not (explicitly) account for other uncertainties related to, for example, rainfall or model skill in reproducing the physics. We have addressed this as a discussion element throughout the revised version of the manuscript (L265, L613-615, L643-644).
- Regarding the universally applied historical error statistics vs a flow-dependent error sampling, we agree with the reviewer and have emphasized this as a limitation in our simplified method in the discussion (L605-613).

The second main comment concerns the rainfall estimation. The IPET approach seems fairly crude, and was shown to severely underpredict rainfall for Idai. There are other computationally efficient TC rainfall risk models in the literature that appear to perform better (e.g., Lu et al. 2018). I would suggest to add additional justification for your choice of IPET and what that means for your results: Lu, P., Lin, N., Emanuel, K., Chavas, D. and Smith, J., 2018. Assessing hurricane rainfall mechanisms using a physics-based model: Hurricanes Isabel (2003) and Irene (2011). Journal of the Atmospheric Sciences, 75(7), pp.2337-2358.

We understand this comment for Reviewer #2 and provided additional justification for our choice of IPET (L630-631), what that means for the results (L632-634), and acknowledgment of other methods such as Lu et al. (L634-636).

For your case study, I would be interested in seeing results of an ensemble subset without the rainfall component to understand it's contribution to compound flooding. You point to multiple flood peaks and suggest attribution, but you have an opportunity to test this running perhaps a subset of the ensemble without rainfall. I consider this suggestion to be optional.

Thank you for the suggestion regarding an analysis without the rainfall component. However, this is quite an effort to rerun the entire analysis (>15 days of simulation time) and we feel this falls outside the scope of our current study, which is primarily focused on introducing and applying the TC-FF methodology. We appreciate the idea and feel this could be a valuable direction for future research. For comparison, we did run the best track from cyclone Idai with

and without rain resulting in the following time series with and without rain. One can see clearly the importance of both compound flooding at the City of Beira but also the lack of influence of tide and storm surge further upstream in the estuary.

[Figure]

Minor comments:

- Please consider citing Fossell et al. (2017). This work provides good motivation for your choice of uncertain track parameters to pertub: Fossell, K.R., Ahijevych, D., Morss, R.E., Snyder, C. and Davis, C., 2017. The practical predictability of storm tide from tropical cyclones in the Gulf of Mexico. Monthly Weather Review, 145(12), pp.5103-5121.

We appreciate this suggestion from Reviewer #2 and added this in the revised manuscript (L140).

- Figure 7: Can you check that the Saffir Simpson category lines and the track intensity all have the same definition (i.e., all 1-minute or all 10-minute sustained values). It looks like they may be different.

We double-checked the Saffir Simpson category lines and noticed they were 1-minute based instead of 10 minutes as all the model results were. We have revised Figure 7 and clarified the conversion (L255-256).

- Line 266: Is the word 'surge' missing near 'resulting storm'

Yes, thanks for noticing, we addressed this in the revised manuscript (L280).

- Line 426: The statement 'TC-FF has no bias corrections' is not strictly true given that rainfall was heavily bias corrected for the case of Cyclone Idai.

We have removed this statement (L451) and revised this statement in the manuscript (L454-455) to focus on the cross-track, along-track, and intensity error.

- I noticed that the citation Done et al (2020) is missing from the reference list.

We added this reference to the list.

- While the intention of the forecasting system is stated as informing operational short-term risk decisions, I also see application to long-term risk management. The large ensemble of 'didn't happen but could have happened' simulations could be used as counterfactuals to support longer-term risk mitigation.  See, for example, Rye and Boyd (2022). You may like to mention this in the discussion: Rye, C.J. and Boyd, J.A., 2022.

Downward Counterfactual Analysis in Insurance Tropical Cyclone Models: A Miami Case Study. In Hurricane Risk in a Changing Climate (pp. 207-232). Cham: Springer International Publishing.

We agree with Reviewer #2 and have added this to the revised manuscript (L652-654).